# Evaluation of the Horizontal Transmission of White Spot Syndrome Virus for Whiteleg Shrimp (*Litopenaeus vannamei*) Based on the Disease Severity Grade and Viral Shedding Rate

**DOI:** 10.3390/ani13101676

**Published:** 2023-05-18

**Authors:** Min-Jae Kim, Jae-Ok Kim, Gwang-Il Jang, Mun-Gyeong Kwon, Kwang-Il Kim

**Affiliations:** 1Department of Aquatic Life Medicine, Pukyong National University, Busan 48513, Republic of Korea; manutd5274@pukyong.ac.kr; 2Tongyeong Regional Office, National Fishery Products Quality Management Service (NFQS), Tongyeong 53019, Republic of Korea; kimjaeok@korea.kr; 3Aquatic Disease Control Division, National Fishery Products Quality Management Service (NFQS), Busan 49111, Republic of Korea; gijang2@korea.kr (G.-I.J.); mgkwon@korea.kr (M.-G.K.)

**Keywords:** white spot syndrome virus, waterborne transmission, severity grade, viral shedding rate, minimum infective dose

## Abstract

**Simple Summary:**

White spot syndrome virus, which causes white spot disease, is the most prevalent crustacean pathogen. However, its waterborne transmission based on the correlation between disease severity grade and viral shedding rate has not yet been explored. In this study, we investigated the horizontal transmission model of white spot syndrome virus based on the correlation between disease severity grade and viral shedding rate. We also determined the minimum infective doses of white spot syndrome virus via the waterborne route. We found a positive correlation between the disease severity grade and viral shedding rate of infected shrimp, suggesting that waterborne transmission of white spot syndrome virus is influenced by the viral load and exposure period. Our results can aid in understanding the dynamics of white spot syndrome virus in pond culture systems and facilitate further studies on the interactions between species during outbreaks and the spread of white spot disease in pond culture systems.

**Abstract:**

White spot syndrome virus (WSSV) is the most problematic pathogen in crustaceans. In this study, we investigated the horizontal transmission model of WSSV based on the correlation between the disease severity grade and viral shedding rate and determined the minimum infective dose of WSSV via the waterborne route. Intramuscular injection challenges at different doses and water temperatures revealed that the thresholds of viral shedding and mortality were G1 (3.1 × 10^3^ copies/mg) and G2 (8.5 × 10^4^ copies/mg), respectively. Furthermore, a positive linear correlation was observed between viral copies of pleopods and viral shedding rate (y = 0.7076x + 1.414; *p* < 0.001). Minimum infective doses of WSSV were determined via an immersion challenge. Infection was observed within 1, 3, and 7 d in 10^5^-, 10^3^-, and 10^1^ copies/mL of seawater, respectively. In the cohabitation challenge, infection was observed within six days with viral loads of 10^1^ to 10^2^ copies/mL of seawater, which further increased in the recipient group. Our results indicate a positive correlation between disease severity grade and viral shedding rate of infected shrimp and suggest that the waterborne transmission of WSSV depends on the viral load and exposure period.

## 1. Introduction

White spot syndrome virus (WSSV), which causes white spot disease (WSD), is the most important crustacean pathogen [1,2]. Since its first report in Taiwan in 1992, WSSV has gradually spread to other regions worldwide [3]. Diseases caused by WSSV result in mass mortality of penaeid shrimp within 10 d of infection and cause substantial economic losses in the shrimp culture industry [4,5].

Similar to most pathogens in invertebrates, the waterborne transmission route is one of the most important horizontal transmission mechanisms for WSSV, along with cannibalism [6]. Therefore, understanding the waterborne transmission model is crucial to preventing the outbreak and spread of diseases [7]. Notably, viral shedding and minimum infective dose are important parameters for estimating and predicting viral loads in seawater, determining the exposure levels for naïve individuals, and understanding the waterborne transmission mechanism of aquatic animal diseases [8,9]. Several studies have investigated the waterborne transmission of viral hemorrhagic septicemia virus (VHSV), infectious hematopoietic necrosis virus, nervous necrosis virus, and red sea iridovirus based on the viral shedding rate of infected fish and the minimum infective dose of the virus [7,10,11,12]. Durand and Lightner [13] determined the minimum infective dose of WSSV via 4 h immersion challenges and reported that viral loads below 10^5^ WSSV genome copies/mL are insufficient to induce infection in whiteleg shrimp (*Litopenaeus vannamei*). In contrast, Kumar et al. [14] and Qayoom et al. [15] reported that continuous exposure to 10^3^ WSSV genome copies/mL in seawater is sufficient to induce infection. Due to the absence of a specific cell line, determination of the minimum infective dose of WSSV under natural conditions is difficult [16].

Severity grades of infectious diseases provide important information about the potential impact of the disease on aquaculture farms. Lightner [17] established the severity grade of WSSV based on the ratio (%) of intracellular inclusion bodies determined via histopathological analysis. Although histopathological findings can reflect the pathological state of host cells, histopathological grading is not highly quantitative and may vary among different diagnostic technicians [18]. Several studies have investigated the severity of WSSV infection using quantitative analyses, including competitive [18] and SYBR green-based real-time [19,20,21] polymerase chain reaction (PCR) assays. Although several studies have established the severity grade of WSSV and investigated the minimum infective dose via immersion challenges, no studies have investigated waterborne transmission based on the correlation between disease severity grade and viral shedding rate.

In this study, we investigated the correlations among clinical changes, disease severity grade, and viral copies of pleopod WSSV-infected shrimp using artificial WSSV-infected whiteleg shrimp at different temperatures. To determine the minimum infective dose of WSSV via the waterborne route, we performed immersion challenges at different administration doses and exposure periods. Moreover, we verified the minimum infective dose of WSSV and its waterborne transmission dynamics based on viral loads in shrimp and seawater via cohabitation challenge.

## 2. Materials and Methods

### 2.1. Shrimp and Virus

Juvenile whiteleg shrimp (2.03 ± 0.85 g) were obtained from an aqua-farm in Geoje, Korea, and they were confirmed to be WSSV-free using nested PCR as previously described by Lo et al. [22] (Appendix A). The shrimp were acclimated in a 250 L aqua tank at 25 ± 0.5 °C for 1 week and fed a commercial diet once daily. The virus used in the present study was extracted from diseased whiteleg shrimp in Taean, Korea in 2014 (WSSV-Te-14 isolate) and purified as previously described [23]. Briefly, WSSV-positive tissue (pleopods, gills, and muscles) was homogenized in a 1:10 solution of phosphate-buffered saline (PBS). The homogenized tissue was centrifuged at 3000× *g* for 30 min, and the supernatant was stored at −80 °C after the filtration process. To preserve the viability of the virus, WSSV was propagated and purified via injection into WSSV-free shrimp before being used in experiments (within 7 days before experimental infection).

### 2.2. Determination of WSSV Genome Copies in Whiteleg Shrimp

To determine WSSV genome copies in whiteleg shrimp, total DNA was extracted from 10 mg of pleopod tissue using a High-Pure PCR Template Preparation Kit (Roche). For quantitative analysis, real-time PCR was performed according to the methods described by Durand and Lightner [13]. Briefly, each 20 μL real-time quantitative polymerase chain reaction mixture contained 1 μL of DNA, which was extracted using the High Pure PCR Template Preparation Kit (Roche, Mannheim, Germany), 300 nM of forward and reverse primers, 150 nM probe (Appendix A), 10 μL of TaqMan™ Universal Master Mix II with UNG (Thermo Fisher Scientific, Waltham, MA, USA), and 7.5 μL of nuclease-free water. Amplification was performed using the StepOne Real-Time PCR System (Applied Biosystems, Foster City, CA, USA). The cycling conditions were as follows: 50 °C for 2 min and 95 °C for 10 min, followed by 40 cycles at 95 °C for 15 s (denaturation) and 60 °C for 1 min (annealing and extension). Quantitative data below the cut-off value, determined by the LOD_95%_ (lower limit of confidence level: 7.24 WSSV genome copies/µL) using serially diluted WSSV-encoded plasmid DNA, were excluded from analysis in this study according to our previous study [23]. Viral genome copies per milligram of pleopod were determined using the following formula:$$\mathrm{WSSV}\;\mathrm{genome}\;\mathrm{copies}/\mathrm{mg}=\frac{\left(\mathrm{WSSV}\;\mathrm{genome}\;\mathrm{copies}\;\mathrm{per}\;µL\;\mathrm{of}\;\mathrm{extracted}\;\mathrm{total}\;\mathrm{DNA}\right)\times \left(\mathrm{volume}\;\mathrm{of}\;\mathrm{extracted}\;\mathrm{total}\;\mathrm{DNA}\right)}{\mathrm{weight}\;\mathrm{of}\;\mathrm{pleopod}\;\mathrm{tissues}\;\mathrm{used}\;\mathrm{to}\;\mathrm{extract}\;\mathrm{the}\;\mathrm{total}\;\mathrm{DNA}\;\left(\mathrm{mg}\right)}$$

### 2.3. Iron Flocculation Assay to Determine WSSV Concentration in Seawater

To concentrate the WSSV particles in the rearing seawater, the PC-ascorbate method of the iron flocculation assay was performed according to a previous study [23]. Briefly, rearing seawater was pre-filtered using a glass microfiber filter (pore size, 1.6 µm; Whatman, Maidstone, UK) and an MF-Millipore™ membrane filter (pore size, 0.45 µm; Merck, Darmstadt, Germany) by a vacuum pump (Gast, Benton Harbor, MI, USA). Next, 50 μL of FeCl_3_ solution (4.83 g per 100 mL of FeCl_3_∙6H_2_O distilled water; final FeCl_3_ concentration of 4.83 μg/mL in seawater) was added to the pre-filtered rearing seawater (500 mL) and dissolved using a magnetic stirrer (<120 rpm; Sigma-Aldrich, St. Louis, MO, USA) for 1 h at room temperature (approximately 25 °C). The Fe-WSSV flocculates were collected by filtering through a polycarbonate membrane (pore size: 1.0 µm; Whatman) using a peristaltic pump at <15 psi (Eyela, Tokyo, Japan). Then, the membrane as Fe-WSSV flocculate was transferred to a 5 mL round-bottom tube, followed by the addition of 1 mL of elution buffer (0.1 M Tris-0.1 M ethylenediaminetetraacetic acid 6H_2_O-0.2 M MgCl_2_∙6H_2_O-0.2 M L-ascorbate, pH 6.0; Sigma-Aldrich). Finally, viral resuspension was performed overnight (approximately 20 h) via a Bio RS-24 Mini-Rotator (30 rpm; Biosan, Riga, Latvia) in a dark room at 4 °C overnight. To determine the number of viral genome copies in the rearing water concentrate, total DNA was extracted from 200 μL of the viral resuspended buffer using a High Pure PCR Template Preparation Kit (Roche), and quantitative analysis was performed using real-time PCR as described above (Section 2.2). The viral shedding rate from WSSV-infected whiteleg shrimp is presented as viral genome copies/L/g/day.

### 2.4. Correlation between Disease Severity Grade and Viral Shedding Rate in WSSV-Infected Shrimp

#### 2.4.1. Intramuscular Injection Challenge

In experiment 1 (Exp. 1), experimental infection of whiteleg shrimp was performed by administration of different doses. The administered doses were selected according to WSSV pathogenicity in juvenile whiteleg shrimp (10^5^ WSSV genome copies/shrimp: induced 100% mortality within 52 h; 10^3^ WSSV genome copies/shrimp: induced infection or moribund state within 52 h; 10^1^ WSSV genome copies/shrimp: unknown dose for inducing infection or mortality) [13,23]. A total of 120 juvenile shrimp were reared in 4 tanks (*n* = 30) and intramuscularly administered with 0.1 mL of diluted WSSV-Te-14 at different doses (10^5^, 10^3^, or 10^1^ WSSV genome copies/shrimp). As a negative control, healthy shrimp (*n* = 30) were intramuscularly administered with 0.1 mL phosphate-buffered saline (PBS; Sigma-Aldrich). Thereafter, shrimp were maintained at 25 °C in 50 L tanks (191.4 individuals/m^2^) for 14 d to observe cumulative mortality, and 500 mL of reared seawater (in triplicate) was collected at 1 and 3 d post-infection (dpi). To obtain time-course samples, a total of 180 whiteleg shrimp from 6 tanks (*n* = 30 per group and sampling days) were collected at 1 and 3 days post-infection (dpi) under identical experimental conditions.

In experiment 2 (Exp. 2), a challenge test was carried out at different temperature conditions (constant [20 °C], constant [30 °C], shifting-up [20 to 30 °C], and shifting-down [30 to 20 °C]). A total of 160 juvenile shrimp (*n* = 20 per group) were acclimated to either 20 or 30 °C for 1 week in 8 20 L aqua tanks and were intramuscularly administered with 0.1 mL of diluted WSSV-Te-14 (10^3^ WSSV genome copies/shrimp). As a negative control, healthy shrimp (*n* = 20) were intramuscularly administered with 0.1 mL PBS. For the shifting-up (20 to 30 °C) group, the temperature of the rearing seawater was increased by 1 °C every 12 h from 1 dpi until it reached 30 °C at 7 dpi. In contrast, for the shifting-down (30 to 20 °C) group, the temperature of the rearing seawater was decreased by 1 °C every 12 h from 1 dpi until it reached 20 °C at 7 dpi. The rearing seawater temperature for the constant (20 °C and 30 °C) groups was maintained throughout the experiment. After experimental infection, shrimp were maintained in 20 L tanks (200.0 individuals/m^2^) for 14 d to observe cumulative mortality, and 500 mL of reared seawater (in triplicate) was collected at 1, 2, and 4 dpi. To obtain time-course samples, a total of 240 whiteleg shrimp in 12 tanks (*n* = 20 per group and sampling days) were collected at 1, 2, and 4 dpi under identical experimental conditions.

During Exps. 1 and 2, the rearing water was replaced once every two days. For quantitative analysis, WSSV particles in the rearing seawater were concentrated using an iron flocculation assay as described above (Section 2.3). Viral copies in challenged shrimp and concentrated WSSV particles were determined using real-time PCR as described in Section 2.2. The results of real-time PCR in the cumulative mortality groups were used to analyze the cut-off value for mortality as described in Section 2.6. The brief schematic diagram of the challenge tests carried out in this study is presented in Appendix A.

#### 2.4.2. Histopathological Analysis and Determination of the Disease Severity Grade

For histopathological analysis, WSSV-infected subcuticular epithelium, gills, hepatopancreas, anterior intestine, and lymphoid organ tissues were processed as previously described [24]. Briefly, WSSV-infected shrimp from the time-course samples in Exp. 1 (*n* = 3 per group) and Exp. 2 (*n* = 5 per group) were randomly selected, and their cephalothoraxes were fixed in Davidson’s solution (Cancer Diagnostics Inc., Durham, NC, USA) for 24 h. Samples were then dehydrated in a graded alcohol series (70–100%) and embedded in paraffin. Tissue sections (4 μm thick) were stained with hematoxylin and eosin (H&E; Abbey Color, Philadelphia, PS, USA). For the negative control, 0.1 mL PBS-administered shrimp in Exp. 1 (*n* = 3 per group) and Exp. 2 (*n* = 5 per group) were fixed, processed, embedded, sectioned, and H&E-stained, similar to the experimental groups. Histopathological analysis and images were obtained using a light microscope (BX50; Olympus Ltd., Tokyo, Japan).

The severity of WSSV infection in the shrimp was determined by assessing the percentage of intracellular inclusion bodies observed through histopathological analysis [17]. In brief, infections with an inclusion body ratio of less than 10% of the total cells were classified as G1, 30–40% as G2, 40–50% as G3, and over 80% as G4. Any negative results were labeled as G0 (Table 1) (Figure 1). A total of 164 shrimp and 6 sections per individual (2 paraffin blocks with 3 sections per block) were used to assess the severity grades, and the severity grades were determined based on the average score of the 6 sections.

#### 2.4.3. WSSV VP28 Gene Expression Analysis

To compare the WSSV propagation levels at different temperatures, WSSV VP28 gene expression was analyzed using a StepOne Real-time PCR system (Applied Biosystems) according to previous studies [25,26]. Briefly, total RNA was extracted from 50 mg of muscle tissue from the time-course sampled shrimp (Exp. 2; *n* = 5 per group) using the yesR™ Total RNA Extraction Mini Kit (GenesGen, Busan, Republic of Korea), and RNA was reverse-transcribed using the PrimeScript™ 1st cDNA Synthesis Kit (Takara, Japan) following the manufacturer’s protocols. The fold difference of VP28 gene expression relative to β-actin using specific primer sets (Appendix A) was determined by the 2^−ΔΔCt^ method as described by Rao et al. [27].

### 2.5. Waterborne Transmission of WSSV under the Mimicking Natural Conditions

#### 2.5.1. Determination of the Minimum Infective Dose via the Waterborne Route

In Exps. 3 and 4, to investigate the minimum infective dose of WSSV via the waterborne route, immersion challenges were performed at different doses and exposure periods. The administered doses were selected based on previous studies (10^5^ WSSV genome copies/mL: induced infection via immersion challenge; 10^3^ WSSV genome copies/mL: induced infection via continual exposure; 10^1^ WSSV genome copies/mL: viral loads in the outlet water of the WSSV outbreak shrimp farm) [13,14,23].

In Exp. 3, eighty juvenile shrimp were reared in four 20 L tanks (*n* = 20) and immersed in WSSV at three different doses (10^5^, 10^3^, or 10^1^ WSSV genome copies/mL) with UV-sterilized seawater at 25 °C for 24 h, and 100% of rearing water was exchanged. As a negative control, healthy shrimp (*n* = 20) were immersed in the PBS-spiked seawater. At 1, 3, 5, 7, 9, and 11 dpi, 500 mL of rearing seawater was collected (in triplicate) to investigate the viral shedding rate of infected shrimp. Under the same experimental conditions, pleopods from whiteleg shrimp (*n* = 3 per group) were collected at 1, 3, 5, 7, 9, and 11 dpi to investigate the time course of viral copies.

In Exp. 4, sixty juvenile shrimp were reared in three 20 L tanks (*n* = 20) and immersed in WSSV at two different doses (10^3^ or 10^1^ WSSV genome copies/mL) with UV-sterilized seawater at 25 °C for 14 d. As a negative control, healthy shrimp (*n* = 20) were immersed in the PBS-spiked seawater. Under the same experimental conditions, pleopods from whiteleg shrimp (*n* = 3 per group) were collected at 1, 3, 5, 7, 9, and 11 dpi to confirm infection.

For the immersion challenges, 0.1 mL of WSSV-Te-14 solution was 10-fold serial-diluted in UV-sterilized seawater to obtain the final challenged dose with a volume of 10 L, and negative control groups immersed in the same volume of PBS-spiked seawater. After immersion challenges, shrimp were maintained at 25 °C in 20 L tanks (200.0 individuals per m^2^) for 14 d to observe cumulative mortality. During the experiment, the rearing water was replaced once every two days. In Exp. 3, UV-sterilized seawater was used as a replacement, whereas in Exp. 4, WSSV-spiked seawater was used at the same initial immersion dose. For quantitative analysis, WSSV particles in the rearing seawater were concentrated using the iron flocculation assay as described above (Section 2.3). Viral copies in challenged shrimp and concentrated WSSV particles were determined using real-time PCR as described in Section 2.2.

#### 2.5.2. Cohabitation Challenge

In Exp. 5, to verify the viral shedding rate of infected shrimp and the minimum infective doses of WSSV, a cohabitation challenge was performed at different WSSV doses and water temperatures. A total of 180 juvenile shrimp (*n* = 20 per group; donor) were acclimated in 9 50 L aqua tanks at 20, 25, or 30 °C for 1 week and intramuscularly administered with 0.1 mL of diluted WSSV-Te-14 at different doses (10^5^ or 10^3^ WSSV genome copies/shrimp). As a negative control, healthy shrimp (*n* = 20) were intramuscularly administered with 0.1 mL PBS. At 3 dpi, plastic cages were installed in each tank to house the healthy shrimp. A total of 180 healthy shrimp (*n* = 20 per group; recipient) were housed in plastic cages in each tank and maintained for 14 d (*n* = 40 per group; 250.0 individuals per m^2^) to observe cumulative mortality. During the experiment, 50% of rearing water was replaced once every two days. Under the same experimental conditions, pleopods from the recipients (*n* = 3 per group) and 500 mL of rearing seawater (in triplicate) were collected at 3, 5, 7, 9, and 11 dpi. Seawater was collected from the tanks in the experimental group to measure mortality. For quantitative analysis, WSSV particles in the rearing seawater were concentrated using an iron flocculation assay as described above (Section 2.3). Viral copies in challenged shrimp and concentrated WSSV particles were determined using real-time PCR as described in Section 2.2.

### 2.6. Statistical Analyses

The statistical significance of viral copies in pleopods, rearing seawater, and VP28 gene expression among the administered groups was determined using two-way ANOVA. Linear regression analysis was performed to analyze the correlation between pleopod viral copies and the viral shedding rate. The cut-off values for viral shedding and mortality of WSSV-infected shrimp were determined using ROC curve analysis. Statistical analyses were conducted using SPSS ver. 27 (IBM) and significance was set at *p* < 0.05.

## 3. Results

### 3.1. Correlation between Viral Copies of Pleopods and Viral Shedding Rate

#### 3.1.1. Cumulative Mortality in Injection-Challenged Groups

In Exp. 1, at different administration doses, the shrimp administered 10^5^ and 10^3^ WSSV genome copies began to die at 1 and 3 dpi, respectively, and exhibited 100.0% cumulative mortality at 4 and 10 dpi, respectively. The experimental group that was injected with 10^1^ WSSV genome copies/shrimp exhibited 6.6% mortality, which was the same as the negative control group (6.6% mortality), and viral genome could not be identified by real-time PCR (Figure 2A).

In Exp. 2, experimental infection was conducted at 20 °C (constant [20 °C] and shifting-up [20 to 30 °C]) and 30 °C (constant [30 °C] and shifting-down [30 to 20 °C]). While the initial mortality of shrimp was observed based on the administered temperature (20 °C: 3 dpi; 30 °C: 2 dpi), the cumulative mortality rates were exhibited differently depending on the temperature shifts (Figure 2B). The cumulative mortality rates of the constant (20 °C) and shifting-up groups were 27.5 and 80.0%, respectively, while that of the constant (30 °C) and shifting-down groups were 47.5% and 87.5%, respectively (Figure 2B).

#### 3.1.2. Viral Shedding Rate in Injection-Challenged Groups

To compare the viral shedding rate among the injection-challenged groups, the viral copies in the pleopod are presented as median (interquartile range [IQR]), and the viral shedding rate is presented as mean ± standard deviation (SD).

In Exp. 1, the viral copies of pleopods from shrimp increased over time (1 and 3 dpi), with significant differences depending on the initial dose administered. The median viral genome copies of the pleopod after injection with 10^5^ and 10^3^ WSSV genome copies/shrimp were 3.1 × 10^5^ (IQR, 1.5 × 10^5^ to 7.3 × 10^5^) and 1.2 × 10^3^ (1.5 × 10^2^ to 7.6 × 10^3^) WSSV genome copies/mg at 1 dpi, and 2.0 × 10^7^ (1.2 × 10^7^ to 2.9 × 10^7^) and 1.2 × 10^6^ (5.4 × 10^5^ to 1.7 × 10^6^) WSSV genome copies/mg at 3 dpi, respectively. Meanwhile, the viral copies in dead shrimp were not significantly different, regardless of the administered dose. The viral shedding rate also increased, with a trend similar to that in the viral copies of pleopods, with a significant difference depending on the initial dose administered (Figure 3A). The mean viral shedding rate of the shrimp administered with 10^5^ and 10^3^ WSSV genome copies/shrimp were 1.7 × 10^4^ (SD, ±9.6 × 10^3^) and 2.6 × 10^2^ (±1.3 × 10^2^) WSSV genome copies/L/g at 1 dpi, and 3.8 × 10^6^ (±1.9 × 10^6^) and 3.1 × 10^5^ (±1.8 × 10^5^) WSSV genome copies/L/g at 3 dpi, respectively (Figure 3A).

In Exp. 2, the viral copies of pleopods from shrimp increased over time (1, 2, and 4 dpi), with significant differences depending on the temperature shift (Figure 3B). The median viral copies of pleopods of shrimp administered at 20 °C (constant [20 °C] and shifting-up) exhibited significant differences at 4 dpi, with 5.8 × 10^3^ (IQR, 2.7 × 10^3^ to 9.0 × 10^3^) and 4.4 × 10^5^ (1.4 × 10^5^ to 1.3 × 10^6^) WSSV genome copies/mg. Furthermore, for the shrimp administered at 30 °C (constant [30 °C] and shifting-down), the median viral genome copies of pleopod showed significant differences at both 2 and 4 dpi, with 3.9 × 10^3^ (3.5 × 10^2^ to 1.8 × 10^4^) and 6.0 × 10^4^ (2.9 × 10^3^ to 5.1 × 10^5^) WSSV genome copies/mg at 2 dpi and 3.5 × 10^4^ (3.1 × 10^3^ to 8.4 × 10^5^) and 1.9 × 10^6^ (2.1 × 10^5^ to 4.9 × 10^7^) WSSV genome copies/mg at 4 dpi. However, the viral copies in dead shrimp showed no significant difference, with median viral copies of pleopods greater than 10^6^ WSSV genome copies/mg, regardless of the water temperature (Figure 3B). The viral shedding rate also increased with a trend similar to that in the viral copies in pleopods, and there was a significant difference depending on the water temperature conditions. The mean viral shedding rate of the shrimp administered at 20 °C (constant [20 °C] and shifting-up) exhibited significant differences at 4 dpi, with 2.9 × 10³ (SD, ±2.1 × 10³) and 2.0 × 10⁶ (±8.3 × 10⁵) WSSV genome copies/L/g, respectively (Figure 4A). Furthermore, for the shrimp administered at 30 °C (constant [30 °C] and shifting-down), the median viral genome copies of pleopods showed significant differences at both 2 and 4 dpi, with 5.4 × 10^3^ (±2.5 × 10^3^) and 5.2 × 10^5^ (±3.2 × 10^5^) WSSV genome copies/L/g at 2 dpi and 1.6 × 10^5^ (±1.2 × 10^5^) and 4.5 × 10^7^ (±2.7 × 10^7^) WSSV genome copies/L/g at 4 dpi, respectively. Notably, viral shedding was not detected or was below the limit of detection at 1 dpi, regardless of water temperature (Figure 4A).

#### 3.1.3. Time-Course WSSV Propagation Levels in Shrimp

In Exp. 2, the viability of WSSV in shrimp (*n* = 5) was determined by measuring the relative expression of VP28, which was normalized to that of the housekeeping gene β-actin (Figure 4B). At 1 dpi, there were no significant difference in VP28 expression among the groups, regardless of water temperature. The shifting-down group exhibited significant upregulation of VP28 gene expression compared to the other groups at 2 dpi. Of note, the relative expression of the VP28 gene was significantly up-regulated in both the shifting-up and shifting-down groups compared to each of the constant (20 °C) and constant (30 °C) groups, respectively, that were administered at the same temperatures (Figure 4B).

#### 3.1.4. Linear Correlation between Viral Copies of Pleopods and Viral Shedding Rate

Based on the viral shedding rates in Exps. 1 and 2, linear regression analysis was carried out to examine the correlation between viral copies of pleopods in shrimp and the viral shedding rate. The analysis revealed a positive correlation (y = 0.7076x + 1.414) (*p* < 0.001) as shown in Figure 4C.

### 3.2. WSSV Severity Grade and Quantitative Analysis

#### 3.2.1. Disease Severity Grade in WSSV-Infected Shrimp

Based on the histopathological findings, the severity grade of the WSSV-infected shrimp was determined according to Lightner [17], and the time-course severity grades are presented in Table 2 and Table 3.

In Exp. 1, the severity grades of WSSV-infected shrimp injected with 10^5^ and 10^3^ WSSV genome copies/shrimp were G2 and G1–G2, respectively, at 1 dpi. At 3 dpi, the severity grades increased to G3–G4 and G2–G3 in shrimp injected with 10^5^ and 10^3^ WSSV genome copies, respectively. In the absence of induced infection, hypertrophic nuclei or pathological lesions were not observed in shrimp after the injection of 10^1^ WSSV genome copies (G0) (Table 2).

In Exp. 2, the severity grade of WSSV-infected shrimp differed depending on the temperature conditions. At 1 dpi, the constant (20 °C) group exhibited G0, while the constant (30 °C) group exhibited G0–G1. Although both groups that were administered at 20 °C (constant [20 °C] and shifting-up) showed G1–G2 severity grades at 2 dpi, due to temperature shifting, the constant (20 °C) and shifting-up groups exhibited G0–G1 and G2–G3, respectively, at 4 dpi. Furthermore, for the shrimp administered at 30 °C (constant [30 °C] and shifting-down) showed different severity grades at both 2 and 4 dpi, with G0–G2 and G1–G2 at 2 dpi and G1–G2 and G3–G4 groups at 4 dpi, respectively. Meanwhile, the surviving shrimp in each group showed G0–G2 severity grades, regardless of the water temperature conditions (Table 3).

#### 3.2.2. Threshold of Clinical Changes in WSSV-Infected Shrimp

The results of the real-time PCR in Exp. 1 and 2 were used to investigate the threshold of clinical changes (viral shedding and mortality) in WSSV-infected shrimp using the cut-off value, which was determined using ROC curve analysis (Figure 5). The cut-off values for viral shedding and mortality were 3.1 × 10^3^ (10^3.49^) and 8.5 × 10^4^ (10^4.93^) WSSV genome copies/mg, respectively (Figure 5).

#### 3.2.3. Correlation between WSSV Severity Grades and Viral Copies of Pleopods

Based on the results of the histopathological analysis and real-time PCR in experiments 1 and 2, The correlation between severity grades and quantitative analysis is presented in Table 4. A total of 23 individuals exhibited a severity grade of G0, with median viral copies of 7.9 × 10^1^ (IQR, 4.3 × 10^1^ to 2.6 × 10^3^) WSSV genome copies/mg observed in the pleopod. Furthermore, the severity grades of G1, G2, and G3 corresponded to median viral copies of 3.1 × 10^3^ (7.4 × 10^2^ to 9.9 × 10^3^), 7.5 × 10^4^ (1.3 × 10^4^ to 5.5 × 10^5^), and 1.9 × 10^6^ (1.4 × 10^5^ to 1.5 × 10^7^) WSSV genome copies/mg, respectively (Table 4). Only four individuals exhibited a severity grade of G4, and the median viral copies of pleopods were exhibited as 1.9 × 10^8^ (6.7 × 10^7^ to 3.2 × 10^8^) WSSV genome copies/mg. Notably, the threshold of viral shedding corresponded to the severity grade of G1, and mortality corresponded to the severity grade of G2 (Figure 6).

### 3.3. Waterborne Transmission of WSSV in Natural-Condition-Mimicking Conditions

#### 3.3.1. Minimum Infective Dose of WSSV via the Waterborne Route

In Exp. 3, at different immersion doses, the shrimp administered 10^5^ WSSV genome copies/mL began to die at 3 dpi and exhibited 65.0% of the cumulative mortality for 14 d. Meanwhile, the shrimp administered with 10^3^ and 10^1^ WSSV genome copies/mL showed no significant difference from the control groups, and viral genomes could not be identified by real-time PCR (Figure 7A). The viral copies of pleopods and viral shedding rate from the shrimp administered 10^5^ WSSV genome copies/mL were first detected at 3 dpi, and both the maximum viral copies of pleopods (7.8 × 10^5^ WSSV genome copies/mg) and viral shedding rate (3.5 × 10^6^ WSSV genome copies/L/g) were observed at 7 dpi (Figure 7A). After 9 dpi, viral copies of pleopods and shedding rate were decline for experimental period (9 and 11 dpi), and the live shrimp in cumulative mortality group exhibited 7.5 × 10^3^ (SD ± 8.2 × 10^3^) WSSV genome copies/mg.

In Exp. 4, the shrimp were immersed in different doses and continually exposed. The cumulative mortality of the shrimp administered with 10^3^ and 10^1^ WSSV genome copies/mL was found to be 85.0 and 35.0%, respectively, over a period of 14 d (Figure 7B). Furthermore, the 10^3^ and 10^1^ WSSV genome copies/mL groups confirmed infection at 3 and 7 dpi, respectively, from the time-course sampled shrimp.

#### 3.3.2. Pathogenicity and Viral Load Dynamics in Cohabitation Challenge

In Exp. 5, the cohabitation challenge was performed with different WSSV administration doses and water temperatures to verify the viral shedding of infected shrimp and minimum infective doses of WSSV under the mimicking natural conditions.

In the groups administered with 10^5^ WSSV genome copies/shrimp, waterborne transmission was successfully induced in the recipient groups at 5 dpi with a cumulative mortality rate above 85.0%, regardless of water temperature. (Figure 8). At 3 dpi, when the recipient groups were housed in the experimental tanks, although 10^4^ to 10^5^ WSSV genome copies/mL of viral loads were present under all temperature conditions, the viral copies of pleopods in the recipients differed depending on the temperature. Similar to the results of injection challenges, the infection progressed most rapidly at 25 °C. In the 20 °C and 30 °C groups, mass mortality of the recipient groups occurred from 7 dpi, with greater than 10^5^ WSSV genome copies/mg of viral copies in pleopods (Figure 9).

In the groups administered with 10^3^ WSSV genome copies/shrimp, 100.0% of cumulative mortality at 25 °C was observed in both the donor and recipient groups. For the shrimp housed at 30 and 20 °C, 40.0 and 30.0% cumulative mortality was observed in the donor groups, while in the recipient groups, 60.0 and 35.0% cumulative mortality were observed, respectively (Figure 8). At 3 dpi, when the recipient groups were accommodated in the experimental tanks, although both the 30 °C and 25 °C groups exhibited approximately 10^3^ WSSV genome copies/mL of viral load in seawater, infections in the recipient groups were observed at 7 and 5 dpi, respectively. For the 20 °C group, the infection of the recipient group was confirmed at 9 dpi, while the viral loads in seawater were exhibited as 10^2^ to 10^3^ WSSV genome copies/mL (Figure 9). Furthermore, the maximum viral copies of pleopods exhibited in the recipient groups coincided with the observation of mass mortality (30 and 25 °C: 9 dpi; 20 °C: 11 dpi; Figure 9).

## 4. Discussion

The viral shedding rate and minimum infective dose of WSSV are some of the most important factors that need to be investigated in waterborne transmission models of aquatic animal pathogens. Furthermore, because WSSV-infected shrimp exhibit various clinical changes (such as viral shedding and mortality) that vary by severity grade [19,20], understanding the correlation between viral shedding and severity grade is important for assessing the potential impact of infected shrimp on naïve individuals. Therefore, this study investigated the correlation among severity grades, viral copies in pleopods, and clinical changes (viral shedding and mortality) of infected shrimp via injection challenges at different administration doses and temperature conditions (Exps. 1–2). Based on the correlation between severity grades and viral shedding rates, we verified the waterborne transmission of WSSV based on minimum infective doses and viral loads in seawater via immersion and cohabitation challenges (Exps. 3–5).

Although the virulence of WSSV varies depending on shrimp species and temperature, penaeid shrimp are generally highly susceptible to WSSV. Therefore, in this study, a horizontal transmission model of WSSV was investigated using whiteleg shrimp, as recommended by the World Organization for Animal Health (WOAH) [28]. In Exp. 1, based on the viral copies determined by real-time PCR, as recommended by the WOAH [28], whiteleg shrimp were administered different doses of WSSV genome (10^5^ as early infection/mortality, 10^3^ as infection/mortality, and 10^1^ as no or low-grade infection). As infection and mortality inducible doses, shrimp administered 10^5^- and 10^3^ WSSV genome copies exhibited 100% cumulative mortality at 4 and 10 dpi, respectively, exhibiting significant differences depending on the initial dose administered (Figure 2A). Similarly, Pang et al. [29] reported that shrimp after administration of 10^4^ WSSV copies had 10^3^ copies/mg of viral copies at 1 dpi and exhibited 100% cumulative mortality in four days. Based on the results of Exp. 1, Exp. 2 carried out a challenge test, administering 10^3^ WSSV genome copies doses at a water temperature range of 20 to 30 °C, which is commonly reported in shrimp culture ponds [23]. Although WSSV induced infection at 20 °C and 30 °C water temperatures, the optimal temperature range for WSSV propagation was reported to be 23 to 28 °C [30]. As in a previous study, the cumulative mortality of WSSV-infected shrimp was significantly increased in the shifting-up (80.0%) and shifting-down (87.5%) groups compared to the constant (20 °C; 27.5%) and constant (30 °C; 47.5%) groups, which had the same administered temperatures (Figure 2B). Although there was a significant difference in the expression of WSSV VP28 gene, which indicates WSSV propagation levels [31], depending on temperature shifts at 4 dpi (Figure 4B), the reduction in mortality of shifting groups after 7 dpi might suggest that the WSSV propagation levels were reduced due to the completion of temperature shifts and the water temperature being outside the optimal range for propagation. Additionally, the viral copies of pleopods exhibited no significant differences between dead and surviving shrimp, regardless of the administered dose and water temperature (Figure 3). Furthermore, although the viral shedding rate also increased with a similar trend to that in the viral copies in pleopods, with a significant difference depending on the initial dose administered and the water temperature, viral shedding could occur at approximately 10^3^ WSSV genome copies/mg (Figure 4). Therefore, we classified the stages of WSSV severity using injection challenge tests (Exps. 1–2) and determined the severity grades of WSSV infection in whiteleg shrimp (juvenile stage) based on histopathological findings.

The severity grades of WSSV-infected whiteleg shrimp were determined according to Lightner [17] based on histopathological findings, and their relatedness to real-time PCR results was investigated. Based on the histopathological findings and the results of real-time PCR of WSSV-infected shrimp, the number of viral copies in the G1 group was 10 to 10^3^ WSSV genome copies/mg. Furthermore, the severity grades of G2, G3, and G4 corresponded to 10^4^ to 10^5^, 10^5^ to 10^7^, and 10^7^ to 10^8^ WSSV genome copies/mg, respectively, in the pleopods (Table 4). Similarly, Tang and Lightner [18] evaluated the correlation between quantitative and histopathological analyses and found that the G2–G3 and G4 severity grades corresponded to 106 and 108 viral copies/mg in infected shrimp, respectively. Although histopathological analysis is not highly quantitative and may vary among diagnostic technicians [18], these results support the correlation between histopathological analysis and real-time PCR (Table 4).

To investigate the correlation between severity grades and clinical changes in WSSV-infected shrimp, we determined the thresholds for viral shedding and mortality as 3.1 × 10^3^ (10^3.49^) and 8.5 × 10^4^ (10^4.93^) WSSV genome copies/mg, respectively, using ROC curve analysis (Figure 5). In Exp. 1, the median number of viral copies in pleopods at the first shrimp death (10^5^ copies/shrimp group at 1 dpi; 10^3^ copies/shrimp group at 3 dpi) was approximately 10^5^ to 10^6^ WSSV genome copies/mg. Meanwhile, in Exp. 2, although the first death occurred at 2–3 dpi (with 10^2^ to 10^3^ WSSV genome copies/mg median viral copies), several individuals in each group showed relatively high loads (approximately 10^4^ to 10^5^ copies/mg) at 2 dpi. (Figure 3). Compared to Sun et al. [32], approximately 10^5^ to 10^6^ WSSV genome copies/mg of viral copies was sufficient to induce mortality in shrimp. Furthermore, the mass mortality of 10^5^ copies/mL immersed shrimp in Exp. 3 (immersion for 24 h) and recipient groups in Exp. 5 (cohabitation challenge) was observed when the mean number of viral copies in the pleopods was greater than approximately 10^5^ copies/mg (estimated severity grade of G2; Figure 7 and Figure 9). There is limited information on WSSV shedding in shrimp, but several previous studies have reported that WSSV can be detected in seawater, even when the number of viral copies in shrimp is approximately 10^2^ to 10^3^ copies/mg [23,31]. These results suggest that the severity grade of G1 (threshold: 3.1 × 10^3^ copies/mg) and G2 (threshold: 8.5 × 10^4^ copies/mg) in shrimp sufficiently induce viral shedding and mortality, respectively (Figure 6). As viability of the virus could not be determined via real-time PCR, further studies are needed to clarify the correlation between histopathological analysis and molecular assays, including a combination of histopathological findings and viral viability determined using viability PCR assays.

Viral shedding is essential for predicting virus loads in seawater and exposure levels for naïve individuals in contact with infected individuals (or exposed to contaminated water) and could be a useful tool for understanding the epidemiology of the disease [7,9]. Therefore, this study analyzed time-course viral loads in pleopods and seawater to investigate the correlation between the infection stages of WSSV and the viral shedding rate. Based on the real-time PCR results for viral copies in pleopods and seawater from Exps. 1 and 2, it was observed that the viral shedding rate increased in a manner similar to the viral copies of pleopods. Furthermore, a linear correlation between viral shedding rate and viral copies in pleopods was confirmed by regression analysis (Figure 4C), indicating a positive relationship between the two variables. In Exp. 2, the minimum viral shedding rate was observed in the constant (20 °C) group at 2 dpi (2.0 × 10^2^ WSSV genome copies/L/g) with G0–G1 severity grades, whereas the maximum viral shedding rate was observed in the shifting-down group at 4 dpi (4.5 × 10^7^ WSSV genome copies/L/g) with G3–G4 severity grades. Furthermore, in Exp. 5, the 103 copies/shrimp-administered groups exhibited maximum viral loads in seawater two days before the observation of mass mortality in the recipient groups (Figure 8 and Figure 9). Several studies have reported a correlation among viral shedding, viral load in host species, and mortality. According to Hershberger et al. [7], peaks in viral shedding in VHSV-infected Pacific herring preceded peaks in daily mortality by 1 to 2 d. In field studies, Jang et al. [33] conducted WSSV surveillance on 20 shrimp culture farms in South Korea and found a positive correlation between viral load in both shrimp and seawater. Moreover, our previous study [23] confirmed that the detection rate of WSSV in shrimp increased in July and September, coinciding with an increase in viral loads in seawater. These findings suggest that the viral shedding rate positively correlates with the severity and viability of the host. Based on these results, the severity grades of shrimp and viral loads in seawater could be useful factors for determining disease severity and tracking disease outbreaks in epidemiological studies.

The minimum infective dose is an important parameter for investigating the risk of horizontal transmission to susceptible species exposed to pathogens [8]. Several epidemiological studies have reported waterborne transmission models based on the minimum infective dose and viral shedding in infected hosts [7,10,21]. Despite several previous studies investigating the minimum infective doses of WSSV via immersion challenges, horizontal transmission of WSSV is influenced by environmental factors (i.e., water temperature and stock density), and owing to the absence of a cell line for culturing WSSV, determination of the minimum infective dose of WSSV is not easily accessible [16,34]. Therefore, to understand the waterborne transmission model of WSSV, we investigated the minimum infective dose of WSSV through immersion challenges at different doses and exposure periods (Exps. 3–4). Thereafter, to verify viral shedding in WSSV-infected shrimp and the minimum infective doses of WSSV, cohabitation challenges (Exp. 5) were carried out at three different temperatures (30, 25, and 20 °C), which are commonly reported in shrimp culture ponds. In Exp. 3, whiteleg shrimp were immersed in 10^5^, 10^3^, and 10^1^ WSSV genome copies/mL seawater for 24 h. Continuous immersion challenges were conducted in Exp. 4 at relatively low doses (10^3^ and 10^1^ WSSV genome copies/mL, respectively) for 14 d (Figure 7). In a previous study, Durand and Lightner [13] found that 10^5^ WSSV genome copies/mL of seawater was sufficient to induce infection and mortality in whiteleg shrimp, while below 10^4^ WSSV genome copies/mL of seawater was insufficient to induce infection via immersion for 4 h. Meanwhile, a previous study reported that post-larval whiteleg shrimp were susceptible to 10^3^ and 10^4^ WSSV genome copies/mL of seawater through immersion challenges with 50% of water exchange conditions [15]. Similarly, in Exp. 3, only the groups exposed to 10^5^ copies/mL of WSSV exhibited infection and mortality, whereas the groups exposed to 10^3^ and 10^1^ WSSV genome copies/mL were confirmed to be infected at 3 and 7 dpi, respectively, under continuous exposure (Figure 7). During the cohabitation challenges of Exp. 5, the viral loads in seawater of the 10^5^ WSSV genome copies/shrimp administration groups were 10^5^ WSSV genome copies/mL, and the recipients were infected within two days, regardless of water temperature (30, 25, and 20 °C; Figure 9). Meanwhile, the 10^3^ WSSV genome copies/shrimp administration groups showed 10^3^ copies/mL of viral load in seawater at 30 and 25 °C, and the infection of the recipients was observed after 4 and 2 days post-accommodation, respectively. Furthermore, in the 20 °C group, the recipients confirmed infection after six days, while the viral loads of seawater were observed at approximately 10^1^ to 10^2^ WSSV genome copies/mL (Figure 9). Based on these results, we found that despite the influence of water temperature on the waterborne transmission of WSSV, 10^5^, 10^3^, and 10^1^ WSSV genome copies/mL in seawater could induce waterborne infections within approximately one, three, and seven days, respectively. These results suggest that waterborne transmission of WSSV is affected not only by the viral load in seawater, but also by the duration of exposure. Therefore, to better understand the waterborne transmission of WSSV, analyzing viral shedding is important for estimating its impact on naïve individuals.

In shrimp pond culture systems, the interaction between infected and naïve hosts could be crucial for sustaining the viability of WSSV [35]. A previous study investigated the viability of WSSV in seawater and reported that WSSV could be preserved for up to 14 d in seawater under laboratory conditions [14]. Song et al. [36] reported that WSSV was consistently detected in shrimp pond seawater from June to September 2007, whereas Kim et al. [23] reported that WSSV was detected for up to 55 d in shrimp pond water during the study period. In Exp. 5 in this study, waterborne transmission was induced in the recipient, and viral loads of seawater were maintained as infection occurred in the recipient, regardless of water temperature (Figure 9). Based on these results, viral loads in seawater could be maintained through interactions with hosts or vectors, and these interactions helped the WSSV remain viable for an extended period. To clarify the horizontal transmission mechanism of WSSV in pond culture systems, further studies are needed to investigate the more complex interactions, such as the influence of cannibalism and the dynamics of WSSV among host species, vectors, and reservoirs.

## 5. Conclusions

In this study, we investigated the correlation between disease severity grade and clinical changes in the WSSV-infected shrimp. Our results revealed that a severity grade of G2 (cut-off value: 10^4.9^ copies/mg) was sufficient to induce mortality in whiteleg shrimp, and the infected shrimp with a severity grade of G1 (cut-off value: 10^3.5^ copies/mg) was able to shed the virus. Based on the correlation between disease severity grade and viral shedding rate, we determined a positive correlation between viral copies of the pleopods and viral shedding rate (y = 0.7076x + 1.414; *p* < 0.001). Furthermore, challenge tests mimicking natural conditions (immersion and cohabitation challenges) revealed that 10^5^, 10^3^, and 10^1^ WSSV genome copies/mL of seawater could induce waterborne infection within 1, 3, and 7 d, respectively. Our findings on the WSSV waterborne transmission model can aid in understanding the dynamics of WSSV in a pond culture system. Moreover, our results can be used to further elucidate the interactions between species during disease outbreaks and the spread of WSD in pond culture systems.

## Figures and Tables

**Figure 1 animals-13-01676-f001:**
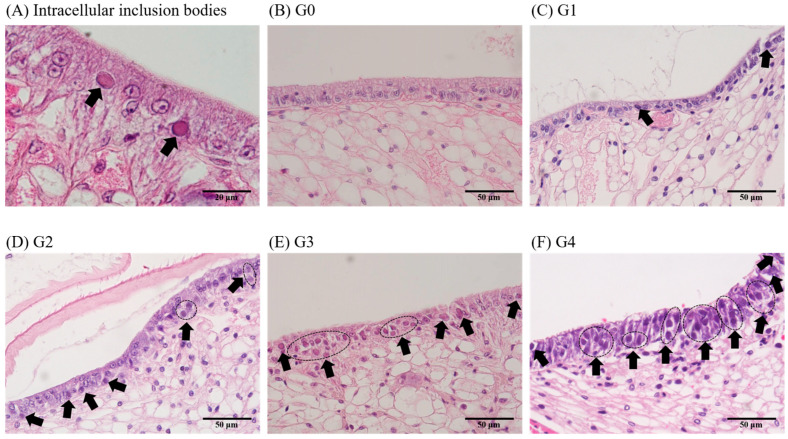
Severity grade of white spot syndrome virus was determined based on the ratio of inclusion bodies (arrow) in the cells of subcuticular epithelium via histopathological analysis.

**Figure 2 animals-13-01676-f002:**
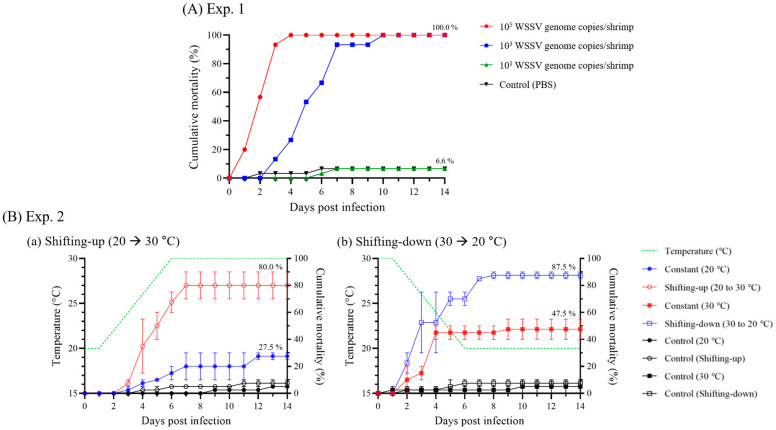
Cumulative mortality due to WSSV experimental infection for 14 d. (**A**) Experiment 1 (*n* = 30/group) performed intramuscular injection using 10^5^, 10^3^, or 10^1^ white spot syndrome virus (WSSV) genome copies/shrimp at 25 °C. (**B**) Experiment 2 (*n* = 20/group) performed intramuscular injection with 10^3^ WSSV genome copies/shrimp under four different temperature conditions: constant (20 °C), shifting-up (20 to 30 °C), constant (30 °C), and shifting-down (30 to 20 °C), with each condition tested in duplicate. The dotted lines indicated the water temperature each day, and the temperature shift was carried out at a rate of 1 °C/12 h. The negative control groups in experiments 1 and 2 were injected with phosphate-buffered saline (PBS).

**Figure 3 animals-13-01676-f003:**
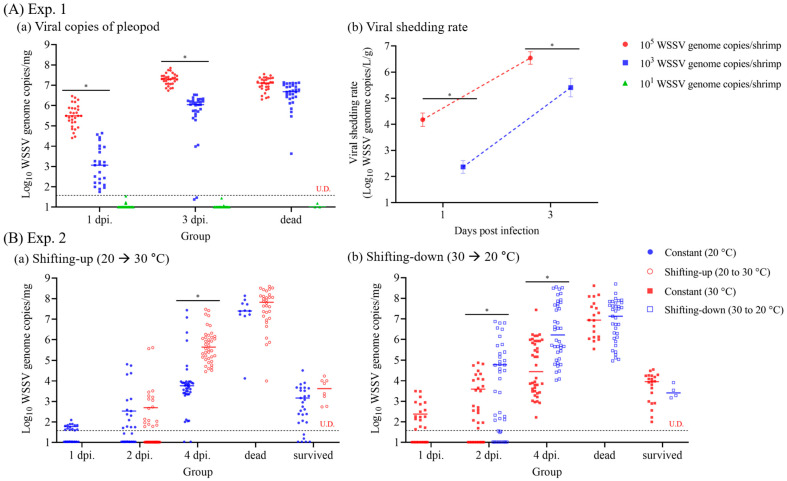
Viral genome copies of the pleopod after intramuscular injection in experiments 1 and 2. (**A**) In experiment 1, whiteleg shrimp (*n* = 30/group) were intramuscular injected with 10^5^, 10^3^, or 10^1^ white spot syndrome virus (WSSV) genome copies/shrimp at 25 °C. Viral genome copies of the pleopod collected at different time intervals (1 and 3 d post-infection and after death). (**B**) In experiment 2, whiteleg shrimp (*n* = 20/group) were intramuscular injection with 10^3^ WSSV genome copies/shrimp under four different temperature conditions (20 °C: shifting-up; 30 °C: shifting-down) at different time intervals (1, 2, and 4 d post-infection, dead and survived individuals). The horizontal bars indicate the median value of each group. The horizontal dotted line indicates the LOD_95%_ value of this study, and U.D. indicates undetermined. Two-way analysis of variance was performed to determine significant differences in viral genome copies of pleopods between experimental groups at *p* < 0.05 (*).

**Figure 4 animals-13-01676-f004:**
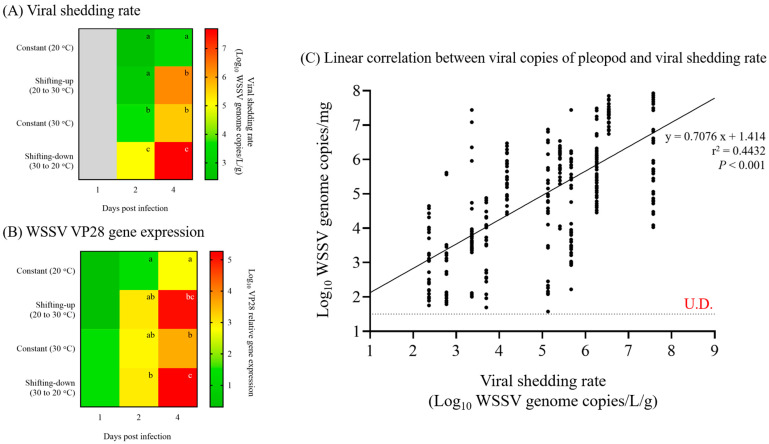
Correlation between viral copies of pleopods and viral shedding rate determined throughout experiment 1 and experiment 2. (**A**) Viral shedding rate (WSSV genome copies/L/g) in Exp. 2 was determined based on the number of viral genome copies in rearing seawater and weight of remaining shrimp. (**B**) WSSV VP28 gene expression in Exp. 2 was determined by the fold difference in VP28 gene expression relative to β-actin by the 2^−ΔΔCt^ method. (**C**) Correlation between viral copies of pleopods and viral shedding rate was determined by linear regression analysis using the results of viral copies of pleopods and viral shedding rate in experiments 1 and 2. The horizontal dotted line indicates the LOD_95%_ value of this study, and U.D. indicates undetermined. Two-way analysis of variance was performed to determine significant differences in viral shedding rate and VP28 gene expression among experimental groups at *p* < 0.05.

**Figure 5 animals-13-01676-f005:**
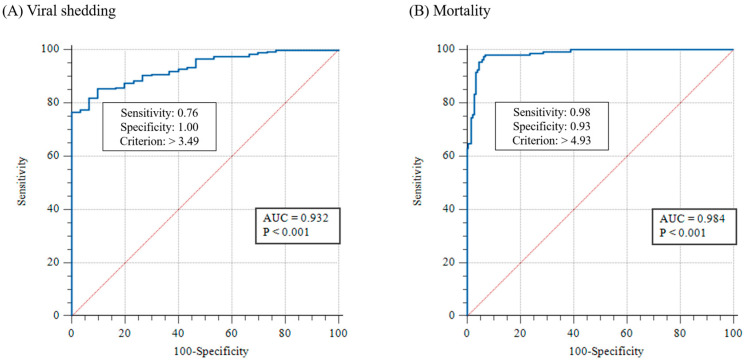
Thresholds for clinical changes of WSSV-infected shrimp, which were determined using ROC curve analysis. The cut-off values of each clinical change were analyzed based on the results of real-time PCR analysis in experiments 1 and 2.

**Figure 6 animals-13-01676-f006:**
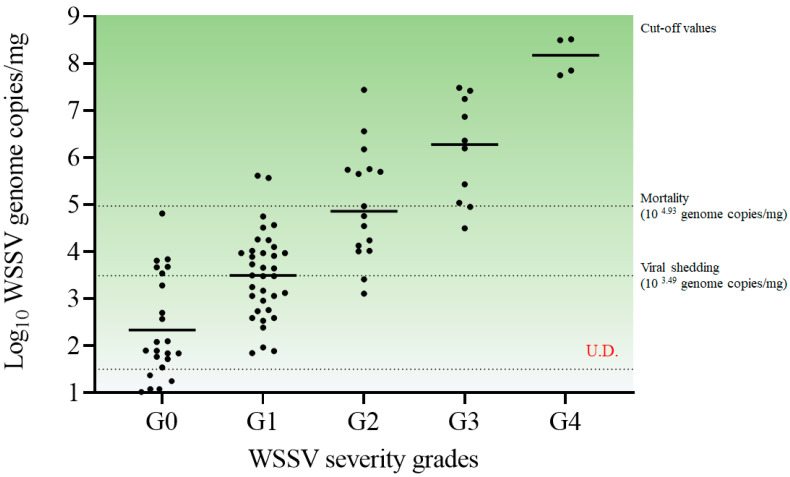
Correlation among severity grades of WSSV, viral copies of pleopods, and thresholds of clinical changes in experiments 1 and 2. Severity grades of WSSV were determined via histopathological findings according to Lightner [17]. The horizontal bars indicate the median value of each group. The horizontal dotted lines indicate the cut-off values of each clinical change or the detection limit of the real-time PCR assay. U.D. indicates undetermined.

**Figure 7 animals-13-01676-f007:**
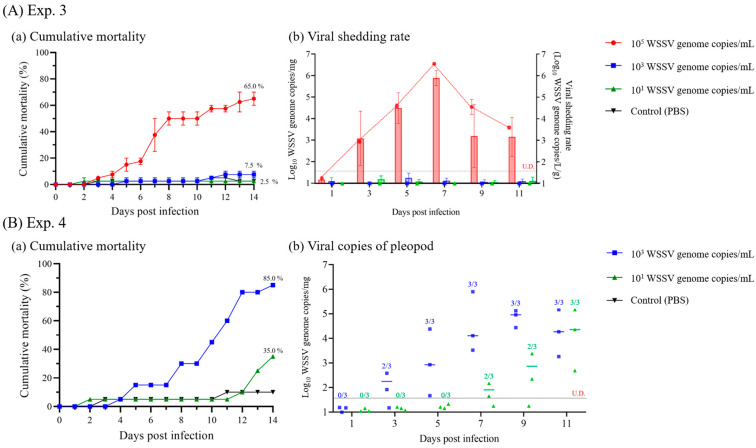
Results of the immersion challenges (experiments 3 and 4). (**A**) In experiment 3, shrimp were immersed with WSSV at three different doses (10^5^, 10^3^ and 10^1^ WSSV genome copies/mL) with UV-sterilized seawater at 25 °C for 24 h. (**a**) Cumulative mortality was observed for 14 d in duplicates. (**b**) Viral shedding rate (WSSV genome copies/L/g) was determined based on the number of viral genome copies in rearing seawater and weight of remaining shrimp. The vertical bars indicate viral copies of pleopods in time-course sampled shrimp (*n* = 3). (**B**) In experiment 4, shrimp were immersed in WSSV-spiked seawater at two different doses (10^3^ and 10^1^ WSSV genome copies/mL) for 24 h. WSSV-spiked seawater was then used to maintain the viral loads in seawater once every two days at the same initial immersion dose. (**a**) Cumulative mortality was observed for 14 d. (**b**) Time-course prevalence (*n* = 3) was determined via real-time PCR with LOD_95%_ value. The number above each scatter plot indicates the result of real-time PCR for the corresponding group. The horizontal dotted line indicates the LOD_95%_ value of this study, and U.D. indicates undetermined.

**Figure 8 animals-13-01676-f008:**
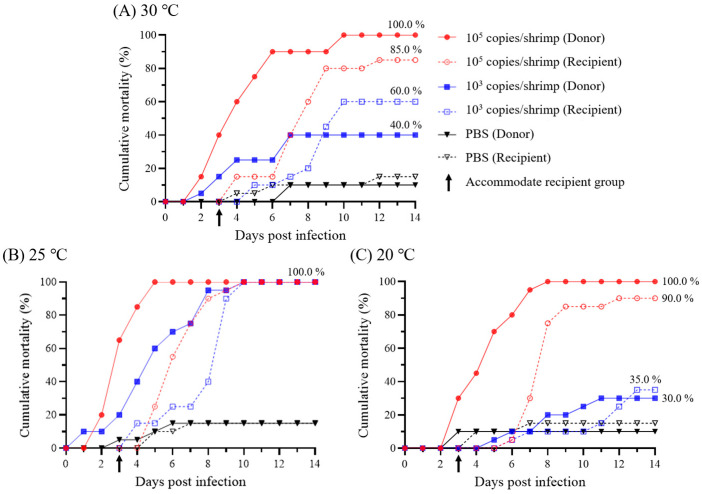
Cumulative mortality of experiment 5 (cohabitation challenge) for 14 d. Whiteleg shrimp were intramuscular injected with 10^5^ or 10^3^ WSSV genome copies/shrimp (Donor; *n* = 20) at 30, 25, or 20 °C. The recipients (*n* = 20) in each group were housed 48 h post-infection and maintained for 14 d in separate cages in the same tank. The negative control groups were injected with phosphate-buffered saline (PBS).

**Figure 9 animals-13-01676-f009:**
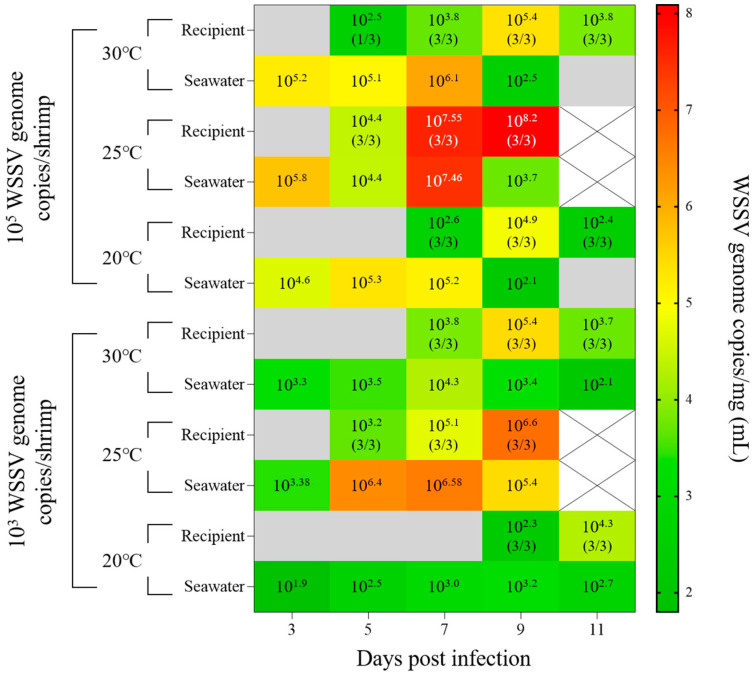
Time-course mean viral copies of pleopods in recipients (*n* = 3) and viral loads of seawater in experiment 5 (cohabitation challenge). The values in each plot indicate the mean viral copies/mg (the prevalence of each sampled shrimp) and mean viral copies/mL. The gray block indicates values below the LOD_95%_ value, and the white ‘x’ marked box indicates time points at which no measurements were carried out due to all shrimp being dead.

**Table 1 animals-13-01676-t001:** Categorization of severity grades of WSSV (Lightner [17]).

Severity Grades of WSSV Infection	Criteria of Categorization
G0	All tissues are normal without any sign
G1	Intracellular inclusion bodies could be seen in less than 10%
G2	Intracellular inclusion bodies could be seen in less than 30–40%
G3	Intracellular inclusion bodies could be seen in less than 40–50%
G4	Intracellular inclusion bodies could be seen greater than 80%

**Table 2 animals-13-01676-t002:** Time-course severity grades of white spot syndrome virus (WSSV) in Exp. 1.

Group		Severity Grade of WSSV ^1^ (*n* = 3)
Administered Dose (WSSV Genome Copies/Shrimp)	Sampling Days	
10^5^	1 dpi ^2^	G2
	3 dpi	G3–G4
10^3^	1 dpi	G1–G2
	3 dpi	G2–G3
10^1^	1 dpi	G0
	3 dpi	G0

^1^ Infection grades of WSSV according to Lightner. (1996); ^2^ dpi, days post-infection.

**Table 3 animals-13-01676-t003:** Time-course severity grades of WSSV in Exp. 2.

Groups	Sampling Days	Severity Grade of WSSV ^1^ (*n* = 5)	Groups	Sampling Days	Severity Grade of WSSV (*n* = 5)
Constant (20 °C)	1 dpi ^2^	G0	Shifting-up (20 to 30)	1 dpi	-
	2 dpi	G0–G1		2 dpi	G0–G1
	4 dpi	G1–G2		4 dpi	G2–G3
	survived	G0–G1		survived	G0–G2
Constant (30 °C)	1 dpi	G0–G1	Shifting-down (30 to 20)	1 dpi	-
	2 dpi	G0–G2		2 dpi	G1–G2
	4 dpi	G1–G2		4 dpi	G3–G4
	survived	G0–G2		survived	G0–G1

^1^ Infection grades of WSSV according to Lightner (1996); ^2^ dpi, days post-infection.

**Table 4 animals-13-01676-t004:** Correlation among WSSV infection grades, viral genome copies, and clinical changes, in WSSV-infected shrimp.

WSSV Severity Grades Based on the Histopathology ^1^	*n* ^ 2^	Viral Genome Copies/mg (Median [IQR ^3^])	Threshold of Clinical Changes in WSSV-Infected Shrimp ^4^
G0	23	7.9 × 10^1^ (4.3 × 10^1^ to 2.6 × 10^3^)	-
G1	35	3.1 × 10^3^ (7.4 × 10^2^ to 9.9 × 10^3^)	Viral shedding (3.1 × 10^3^ WSSV genome copies/mg)
G2	16	7.5 × 10^4^ (1.3 × 10^4^ to 5.5 × 10^5^)	Mortality (8.5 × 10^4^ WSSV genome copies/mg)
G3	10	1.9 × 10^6^ (1.4 × 10^5^ to 1.5 × 10^7^)	-
G4	4	1.9 × 10^8^ (6.7 × 10^7^ to 3.2 × 10^8^)	-

^1^ Infection grades of WSSV according to Lightner [17]; ^2^ the number of individuals corresponding to the severity grade in Exp. 1 and Exp. 2; ^3^ IQR, interquartile range; ^4^ threshold of clinical changes in WSSV-infected shrimp determined using ROC curve analysis.

## Data Availability

Not applicable.

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
