# Peer review of "Evaluation of the Horizontal Transmission of White Spot Syndrome Virus for Whiteleg Shrimp (Litopenaeus vannamei) Based on the Disease Severity Grade and Viral Shedding Rate"

_animals, 2023, doi:10.3390/ani13101676_

Round 1
Reviewer 1 Report
Report on reviewing a manuscript with the title ‘Evaluation of the Horizontal Transmission of White Spot Syndrome Virus Based on the Disease Severity Grade and Viral Shedding Rate’
The current work studies the horizontal infection of white spot syndrome virus in whiteleg shrimp. The paper is well-written, the methods are described properly and the results will contribute to increasing the knowledge repertoire on the studied topic and will open the door to similar work on other pathogens in aquaculture.
Title:
Please, specify your work to the shrimp you used with the Latin name of it included!
Materials and methods
- Even though the quantification of the virus is well explained in the reference used, it is still necessary to refer to the method briefly in your work. Readers can further look for the details in the reference used if they want.
- Line 127: Please, change ‘artificial infection’ to ‘ experimental infection’
- Line 128: please remove this sentence ‘which could or could not induce infection or mortality’.
- Line 128: Please, mention the doses used in brackets.
- The description of the experiment might be easier to follow if the author use a simple sketch to explain what has been done in the five infection trials, including all details.
- In line 137: the authors mentioned ‘Under the same experimental conditions’. Does this mean that two experiments were run in parallel, one for sampling and one for mortality calculation? Please, make it clear!
- Line 147 shifting down (from 30 to 20) not up.
- Was the mortality observed in the same tanks the samples were collected from or was that done in different tanks?
- Line 155: use one way of writing numbers ‘one and two’ or ‘1 and 2’. Please consider this throughout the text!
- If the feeding conditions mentioned in line 156 are not different elsewhere, please mention it in the first paragraph in M&M where you described the animals and the virus. This should be enough, not to distract the readers by redundant info.
- Lines 159 -163: ‘The viral 159 copies in the pleopod are presented as median (interquartile range [IQR]), and the viral 160 shedding rate is presented as mean ± standard deviation (SD)’ should be moved to the result section.
- Please reformulate lines 177 – 179. it was difficult for me to get what you want to say.
- Please make a simple table to describe in detail the grading criteria shown in lines (180 – 182) and attach it to Figure 1 and remove it from the caption of Figure 1.
- Line 179, what ratio? The ratio should be between two variables. Please make it clear!
- Was the number of inclusion bodies determined by the naked eye? It is important then to mention how many fish, sections, and fields have been examined to judge the severity in this study.
- Were the sections analyzed in the scoring experiment chosen based on the phenotypic criteria of the disease (white spots on the external surface of the shrimp) or randomly??
- Line 192: viability or virulence? Please check the function of VP28 gene/protein
- Line 195: revise the ‘yesRTM’ is this the name of the kit?
- Line 209: Please, use a small ‘s’ letter in’Sixty’.
- Line 215: ‘In the Exp. 4, forty juvenile shrimp were reared in three 20 L tanks (n = 20)’ Do you mean two tanks, please revise. Also please remove ‘the’ before ‘Exp.4’
- Line 223 and 224: please use either the standard or the word form of number writing. Also, use the same format throughout the manuscript.
- Please elaborate, how the UV-sterilized/ spiked water was ude in Exp-.3 and Exp.4, respectively ? Was all the water removed and then new water was added, or what? If not, then how much of the water was changed? Also, how did you avoid the additional effect of the virus shedding from the infected shrimp in the Exp.4, 10-3 WSSV DNA copies/mL must have been changed during the days with no water replacement? In the figure 7B, there is no viral shedding rate to judge this.
- Line 225: What the authors mean by PBS solution 10-5
- line 233: the author mentioned (n = 20 per group), even though no groups have been identified before. Please make the number written as ‘n/tank’ and determine which groups have been allocated to which treatment, the sketch I asked for above in the comments might solve this confusion.
- Also, in all the experiments the author did not consider the number of the control fish (pbs injected or immersed) in the total number of the fish. Please fix this by mentioning how many fish (in total including the control groups) have been used in each experiment, how many tanks, what are the group numbers and what are the group identity.
Results:
- Line 260: ‘…. and exhibited 100.0% cumulative mortality’ At what time point this was reported? Please make this clear in the text.
- Line 265: what do the authors mean by ‘ the first mortality’? Do you mean the first notice of mortality?
- Lines 264 – 266, please reformulate. It is confusing as it is right now.
- Line 269: ‘…… were 80.0 and 87.5% cumulative mortality…’ please change ‘ were’ to ‘ had’.
- I am wondering how you comment on the difference in cumulative mortality between the constant 30 and shifting up to 30 groups. The shrimp spent almost the same time 8 days (constant 30) and 10 (shifting-up to 30 C group), however, the cumulative mortality is significantly different between the two groups. Please, also refer to this in the discussion!
- Line 271: Please change to ‘Cumulative mortality due to WWS experimental infection
- Line 272: Please, change to intramuscular injection was performed using …..
- Line 273: I am not sure if ‘DNA copies’ is correct. Please use just copies and remove DNA. Also, consider this throughout the manuscript.
- In Figure 3 what U.D means, please make it clear in the caption.
- Line 283: the authors mentioned that ‘……and 1.2 × 103 (1.5 × 102 to 7.6 × 103 )’. The calculation of log10 of 1200 is 3.07 og However, in Figure 3 the median value is shown to be below 3. Which of the two numbers is wrong? Please check all the other groups, accordingly.
- Line 288 – 292: The results in this paragraph are shown in which figure?
- Line 297: Please move ‘ at 4dpi’ to line 296 after ‘exhibited significant differences ’ to be exhibited significant differences at 4dpi.
- In line 349 and figure 4, Could you please mention the ‘p value’ of correlation?
- Line 380: what the authors mean by ‘Exp. Compounds 1 and 2’
- Line 428: How the infection was confirmed? Please elaborate!
- In the caption of Figure 7, please make the title more clear. F.ex., ‘Results of the immersion challenges of ….shrimp using……pathogen’. Also, make this also clear in the other figures, as the figure caption should be informative on its own.
- Line 443: I am quite against the word ‘Natural’. The experiment is still under controlled conditions, with a commercial diet and changing of the water every other day. This cannot be seen as natural. Please reformulate! Maybe something like, under similar aquaculture conditions, or something similar.
- In Figures 4 and 9, what the grey blocks represent? Are they undetected or not tested samples and why it is so in some groups at some time points in Figure 9? Please, make it clear in the caption.
The language needs a quick polishing.
Author Response
Animals
Manuscript ID: ISSN 2076-2615
Title: Evaluation of the Horizontal Transmission of White Spot Syndrome Virus Based on the Disease Severity Grade and Viral Shedding Rate
# Reviewer 1
Report on reviewing a manuscript with the title ‘Evaluation of the Horizontal Transmission of White Spot Syndrome Virus Based on the Disease Severity Grade and Viral Shedding Rate’
- The current work studies the horizontal infection of white spot syndrome virus in whiteleg shrimp. The paper is well-written, the methods are described properly and the results will contribute to increasing the knowledge repertoire on the studied topic and will open the door to similar work on other pathogens in aquaculture.
Title:
Q1. Please, specify your work to the shrimp you used with the Latin name of it included!
A1. Thanks for your valuable comments. According to your suggestion, we revised our manuscript title from 'Evaluation of the horizontal transmission of white spot syndrome virus based on the disease severity grade and viral shedding rate' to 'Evaluation of the horizontal transmission of white spot syndrome virus for whiteleg shrimp (Litopenaeus vannamei) based on the disease severity grade and viral shedding rate' by adding the name of the shrimp species used in this study.
Materials and methods
Q2. Even though the quantification of the virus is well explained in the reference used, it is still necessary to refer to the method briefly in your work. Readers can further look for the details in the reference used if they want.
A2. Thanks for your comments. Accordingly, we revised the method for determining the cut-off value by adding a specific value, as we did not determine the LOD95% value for this study. Instead, we referred to the cut-off value established in our previous study, which was conducted in the same laboratory (Line 105-108).
(Line 105-108) Quantitative data below the cut-off value, determined by the LOD95% (lower limit of confidence level: 7.24 WSSV genome copies/µL) using serially diluted WSSV-encoded pDNA, were excluded from analysis in this study, according to our previous study [23].
Q3. Line 127: Please, change ‘artificial infection’ to ‘experimental infection’
A3. Thank you for your comments. We have changed “artificial infection” to “experimental infection” based on your comment (Line 136).
Q4. Line 128: please remove this sentence ‘which could or could not induce infection or mortality’.
A4. Thank you for your comments. We have removed the sentence “which could or could not induce infection or mortality” according to your comment (Line 138).
Q5. Line 128: Please, mention the doses used in brackets.
A5. Thank you for your comments. Accordingly, we have revised the unit of doses from “copies” to “WSSV genome copies/shrimp” (Line 138-140)
Q6. The description of the experiment might be easier to follow if the author use a simple sketch to explain what has been done in the five infection trials, including all details.
A6. Thanks for your insights. According to your comment, we added a brief schematic diagram of the challenge tests carried out in this study to aid readers' understanding (Line 173-174). However, since this study already contains several figures, we have included the diagram as Supplemental Figure 1.
Q7. In line 137: the authors mentioned ‘Under the same experimental conditions’. Does this mean that two experiments were run in parallel, one for sampling and one for mortality calculation? Please, make it clear!
A7. Thanks for your comment. Based on your suggestion, we have revised the method description of time-course samples for greater clarity (line 145-149; 162-166; 222-225; 228-229).
Q8. Line 147 shifting down (from 30 to 20) not up.
A8. Thanks for your comment. Based on your suggestion, we have revised the temperature exactly (Line 151-153).
Q8. Was the mortality observed in the same tanks the samples were collected from or was that done in different tanks?
A8. Thanks for your comment. Based on your suggestion, we have revised the method description of time-course samples for greater clarity (line 145-149; 161-165; 222-224; 228-229).
Q9. Line 155: use one way of writing numbers ‘one and two’ or ‘1 and 2’. Please consider this throughout the text!
A9. Thank you for your comment. As per your suggestion, we have now standardized the numbering format of the experiments in this study to Arabic numerals throughout.
Q10. If the feeding conditions mentioned in line 156 are not different elsewhere, please mention it in the first paragraph in M&M where you described the animals and the virus. This should be enough, not to distract the readers by redundant info.
A10. Thank you for your comment. As per your suggestion, we have removed the description of feeding frequency from the Materials and Methods section of the experimental infections.
Q11. Lines 159 -163: ‘The viral 159 copies in the pleopod are presented as median (interquartile range [IQR]), and the viral 160 shedding rate is presented as mean ± standard deviation (SD)’ should be moved to the result section.
A11. Thanks for your comment. As per your comment, we have moved the description of how the viral shedding rate for injection-challenged groups is presented to the Results section. (Line 295-297).
Q12. Please reformulate lines 177 – 179. it was difficult for me to get what you want to say.
A12. Thanks for your comment. According to your comment, we have improved the description of severity grades to make it clearer to the best of our abilities (Line 187-194).
(Line 188-195) The severity of WSSV infection in the shrimp was determined by assessing the percentage of intracellular inclusion bodies observed through histopathological analysis [17]. In brief, infections with an inclusion body ratio of less than 10% of the total cells were classified as G1, 30-40% as G2, 40-50% as G3, and over 80% as G4. Any negative results were labeled as G0 (Table 1) (Figure 1). Total 164 shrimps and six sections per individual (2 paraffin blocks with 3 sections per block) were used to assess the severity grades, and the severity grades were determined based on the average score of the 6 sections.
Q13. Please make a simple table to describe in detail the grading criteria shown in lines (180 – 182) and attach it to Figure 1 and remove it from the caption of Figure 1.
A13. Thank you for comment. As per your suggestion, we have revised the legend for Figure 1. Additionally, we added a table to clearly describe the criteria used for categorizing severity grades and assessing severity grades (line 196).
Q14. Line 179, what ratio? The ratio should be between two variables. Please make it clear!
A14. Thanks for your comment. According to your comment, we have revised the description as (Line 189-190).
(Line 189-190) In brief, infections with an inclusion body ratio of less than 10% of the ‘total cells’ were classified as G1, 30-40% as G2, 40-50% as G3, and over 80% as G4.
Q15. Was the number of inclusion bodies determined by the naked eye? It is important then to mention how many fish, sections, and fields have been examined to judge the severity in this study.
A15. Thanks for your comment. Total 164 shrimps and six sections per individual (2 paraffin blocks with 3 sections per block) were used to assess the severity grades, and the severity grades were determined based on the average score of the 6 sections. According to your comment, we have added an explanation about the number of sections used in this study at Line 191-194.
Q16. Were the sections analyzed in the scoring experiment chosen based on the phenotypic criteria of the disease (white spots on the external surface of the shrimp) or randomly??
A16. Thanks for your comment. We randomly selected shrimp from the time-course samples and used them for histopathological analysis. As per your comment, we have added an explanation at Line 178-181.
(Line 177-180) Briefly, WSSV-infected shrimp from the time-course samples in Exp. 1 (n = 3 per group), and Exp. 2 (n = 5 per group) were “randomly” selected and their cephalothorax were fixed in Davidson’s solution (Cancer Diagnostics Inc, Durham, NC, USA) for 24 h.
Q17. Line 192: viability or virulence? Please check the function of VP28 gene/protein
A17. Thanks for your comment. We investigated VP28 gene expression for propagation levels of WSSV at different temperatures. Therefore, we revised “viability of WSSV” to “WSSV propagation levels”. (Line 201-203)
Q18. Line 195: revise the ‘yesRTM’ is this the name of the kit?
A18. Thanks for your comment. ‘yesRTM’ Total RNA Extraction Mini Kit’ is the name of the kit used this study.
Q19. Line 209: Please, use a small ‘s’ letter in’Sixty’.
A19. Thanks for your comment. We have revised the capital 'S' to lowercase 's' in 'sixty'.
Q20. Line 215: ‘In the Exp. 4, forty juvenile shrimp were reared in three 20 L tanks (n = 20)’ Do you mean two tanks, please revise. Also please remove ‘the’ before ‘Exp.4’
A20. Thanks for your comment. Meanwhile we revised out umber of shrimps and tanks including negative control groups, We not revied the number of tanks used in this study correctly (three [our mistake; two tanks] à three [two infection groups and one control group).
Q21. Line 223 and 224: please use either the standard or the word form of number writing. Also, use the same format throughout the manuscript.
A21. Thank you for your comment. As per your suggestion, we have now standardized the numbering format of the experiments in this study to Arabic numerals throughout.
Q22. Please elaborate, how the UV-sterilized/spiked water was ude in Exp-.3 and Exp.4, respectively ? Was all the water removed and then new water was added, or what? If not, then how much of the water was changed?
A22. Thank you for your comment. In Exp. 3, we exchanged 100% of the rearing water with fresh UV-sterilized seawater immediately after the 24-hour immersion challenge to ensure that there was no residual viral particles left in the water. We have added this description at line 218-220 to provide clarity on this aspect of the experiment.
(Line 218-220) In Exp. 3, eighty juvenile shrimps were reared in four 20 L tanks (n = 20) and immersed in WSSV at three different doses (105, 103, or 101 WSSV DNA copies/mL) with UV-sterilized seawater at 25 °C for 24 h and “100% of rearing water was exchanged”.
Q23. Also, how did you avoid the additional effect of the virus shedding from the infected shrimp in the Exp.4, 10-3 WSSV DNA copies/mL must have been changed during the days with no water replacement? In the figure 7B, there is no viral shedding rate to judge this.
A23. Thank you for your comment. Although viral shedding from infected shrimp influenced the viral loads in the rearing seawater, our objective in Exp. 4 was to investigate how long a period needs to induce WSSV-infection at relatively low viral doses (insufficient to induce infection in a short period of exposure but are detectable on a field scale according to previous studies). Additionally, since viral shedding occurred after induced infection, we determined that it might not influence the first point to induced infection in shrimp.
Q24. Line 225: What the authors mean by PBS solution 10-5
A24. Thanks for your comment. According to your comment, we have added an explanation about the method of immersion challenges more clearly (line 219-220; 231-233).
(Line 220-221) As a negative control, healthy shrimp (n = 20) were immersed in the PBS-spiked seawater.
(Line 232-234) For the immersion challenges, 0.1 mL of WSSV-Te-14 solution was 10-fold serial diluted in UV-sterilized seawater to obtain the final challenged dose with a volume of 10 L and “negative control groups immersed in the same volume of PBS-spiked seawater”.
Q25. line 233: the author mentioned (n = 20 per group), even though no groups have been identified before. Please make the number written as ‘n/tank’ and determine which groups have been allocated to which treatment, the sketch I asked for above in the comments might solve this confusion.
A25. Thanks for your insights. According to your comment, we added a brief schematic diagram of the challenge tests carried out in this study to aid readers' understanding (Line 173). However, since this study already contains several figures, we have included the diagram as Supplemental Figure 1. Furthermore, we have added an explanation about the total shrimps per tank for challenge test (Line 251-253).
(Line 251-253) A total of 180 healthy shrimp (n = 20 per group; recipient) were housed in plastic cages in each tank and maintained for 14 d (“n = 40 per group”; 250.0 individuals per m2) to observe cumulative mortality.
Q26. Also, in all the experiments the author did not consider the number of the control fish (pbs injected or immersed) in the total number of the fish. Please fix this by mentioning how many fish (in total including the control groups) have been used in each experiment, how many tanks, what are the group numbers and what are the group identity.
A26. Thanks for your comment. As per your suggestion, we have unified the number of total shrimps in each experiment including negative control groups throughout.
Results:
Q27. Line 260: ‘…. and exhibited 100.0% cumulative mortality’ At what time point this was reported? Please make this clear in the text.
A27. Thank you for your comment. As per your suggestion, we revised the results to include the time point at which 100% cumulative mortality occurred (Line 271-273).
(Line 272-274) In Exp. 1, at different administration doses, the shrimp administered 105 and 103 WSSV DNA copies began to die at 1 and 3 dpi, respectively, and exhibited 100.0% cumulative mortality at 4 and 10 dpi respectively.
Q28. Line 265: what do the authors mean by ‘the first mortality’? Do you mean the first notice of mortality?
Q29. Lines 264 – 266, please reformulate. It is confusing as it is right now.
Q30. Line 269: ‘…… were 80.0 and 87.5% cumulative mortality…’ please change ‘ were’ to ‘ had’.
A28-30. Thanks for your comment. As per your comment, we revised the result to make it clearer to the best of our abilities (Line 278-284).
Q31. I am wondering how you comment on the difference in cumulative mortality between the constant 30 and shifting up to 30 groups. The shrimp spent almost the same time 8 days (constant 30) and 10 (shifting-up to 30 C group), however, the cumulative mortality is significantly different between the two groups. Please, also refer to this in the discussion!
A31. Thanks for your comment. I completely agree your questions and your question could help to improve our manuscript. As you know, temperature is one of the most important environmental factors affecting pathogen propagation. Therefore, we have additionally described the reduction in mortality in the temperature shifting groups after the 7th day post-infection when the temperature had finished shifting (line 534-538).
(Line 534-538) Although there was a significant difference in the expression of WSSV VP28 gene, which indicates WSSV propagation levels [31], depending on temperature shifts at 4 dpi (Figure 4B), the reduction in mortality of shifting groups after 7 dpi might suggest that the WSSV propagation levels were reduced due to the completion of temperature shifts and the water temperature being outside the optimal range for propagation.
Q32. Line 271: Please change to ‘Cumulative mortality due to WWS experimental infection
Q33. Line 272: Please, change to intramuscular injection was performed using …..
A32-33. Thanks for your comments. We revised figure 2 legend as per your suggestions (Line 286-293).
Q34. Line 273: I am not sure if ‘DNA copies’ is correct. Please use just copies and remove DNA. Also, consider this throughout the manuscript.
A34. Thanks for your comment. As per your suggestion, we have unified the unit of viral copies to ‘WSSV genome copies’ throughout.
Q35. In Figure 3 what U.D means, please make it clear in the caption.
A35. Thanks for your insight. Accordingly, we added the description about the mean of ‘U.D.’ (Line 343-344) The horizontal dotted line indicates the LOD95% value of this study, and U.D. indicates undetermined.
Q36. Line 283: the authors mentioned that ‘……and 1.2 × 103 (1.5 × 102 to 7.6 × 103)’. The calculation of log10 of 1200 is 3.07 og However, in Figure 3 the median value is shown to be below 3. Which of the two numbers is wrong? Please check all the other groups, accordingly.
A36. Thanks for your insight. Since we also presented values below the detection limit in the graph, the position of the bars appeared to shift to accommodate those values. We have reviewed and revised the figure to accurately represent the data.
Q37. Line 288 – 292: The results in this paragraph are shown in which figure?
A37, Thanks for your comment. As per your comment, we added the result of viral shedding rate of Exp. 1 in figure 1A.
Q38. Line 297: Please move ‘at 4dpi’ to line 296 after ‘exhibited significant differences’ to be exhibited significant differences at 4dpi.
A38. Thanks for your comment. We move ‘at 4 dpi’ as per your comment (Line 325-328).
(Line 325-328) The mean viral shedding rate of the shrimps administered at 20 °C (Constant [20 °C] and Shifting-up) exhibited significant differences at 4 dpi, with 2.9 × 10³ (SD, ± 2.1 × 10³) and 2.0 × 10⁶ (± 8.3 × 10⁵) WSSV genome copies/L/g, respectively (Figure 4A).
Q39. In line 349 and figure 4, Could you please mention the ‘p value’ of correlation?
A39. Thanks for your comment. As per your comment, we added the p value of the linear regression at line 371-372 and figure 4C.
Q40. Line 380: what the authors mean by ‘Exp. Compounds 1 and 2’
A40. Thanks for your comment. As per your comment, remove this word ‘Compounds’.
Q41. Line 428: How the infection was confirmed? Please elaborate!
A41. Thanks for your comment. As per your comment, we added the result of real-time PCR in Figure 7B.
Q42. In the caption of Figure 7, please make the title more clear. F.ex., ‘Results of the immersion challenges of ….shrimp using……pathogen’. Also, make this also clear in the other figures, as the figure caption should be informative on its own.
A42. Thank you for your comment. Following your suggestion, we have carefully reviewed and revised the figure legends not only for Figure 7 but also throughout our manuscript. We believe that the revised versions are clearer and more accurate. All the changes made to the figure legends have been marked in red for easier identification.
Q43. Line 443: I am quite against the word ‘Natural’. The experiment is still under controlled conditions, with a commercial diet and changing of the water every other day. This cannot be seen as natural. Please reformulate! Maybe something like, under similar aquaculture conditions, or something similar.
A43. Thanks for your insight. As per your comment, we revised the word ‘natural’ to ‘mimicking natural’ (Lime 463-465).
Q44. In Figures 4 and 9, what the grey blocks represent? Are they undetected or not tested samples and why it is so in some groups at some time points in Figure 9? Please, make it clear in the caption.
A44. Thank you for your insight. The gray blocks represent negative or data below the cut-off values. Additionally, we believe it is necessary to distinguish between values that were not detected and time points at which no measurements were carried out due to all shrimp being dead. Thus, we have indicated the latter group with a white block and a 'x' mark and have revised the figure legend accordingly (Line 498-500).

Reviewer 2 Report
☆ First, I will note what I noticed in the argument leading to the conclusion.
Although the results were obtained in a laboratory setting, the conclusions are reminiscent of the natural occurrence of the pathogen through water in an aquaculture farm. This experiment does not represent a final conclusion on the pathogenicity of WSSV. Therefore, the results are based on the assumption that the virus tested is 100% active. It would be difficult to obtain analytical results if there were conditions that would reduce titer, such as frozen storage. We hope that in the future it will be possible to determine the risk of infection by quantifying the active virus, although this is not necessarily the case with the paper submitted here. Also, the effect of cannibalism of eating dead shrimp was not mentioned, but even if authors are in a position to discuss only waterborne infections, it needs to be included in the discussion.
In the experiment in Figure 4, the weight of the shrimp appears to be relevant, but there is no mention of the weight of the shrimp tested. In the series of experiments, the size of the shrimp (weight, age in days, etc.) seems to be an important factor, but it is not mentioned. Were the series of experiments conducted with shrimp of similar size?
☆ Below are some minor revises
line :86
・Should shortly indicate the specific method of acquiring the virus source.
line:101-102
・It is the weight of the tissue of the pleopod used for the extraction of total DNA, but is it not appropriate to divide it by the amount of the total DNA extracted?
line:127
・artificial infection - replace with experimental infection
line:316
・No explanation for "*" in Figure 3.
line:346
・It is reasonable to only observe a trend of positive correlation due to the low r-value.
line:379
From Figure 5, the cutoff values for viral shedding and cumulative mortality are 3.1 x 103 (103.49) and 8.7 x 104 (104.97) WSSV DNA copies/mg, respectively, but isn't the mortality 10^4.93? It would be easier to understand if more emphasis is placed on the Criterion in the figure.
line:402
・ Table 3. It would be easier to understand if the values of 10^3.49 and 10^4.93 or 4.97 are also included in the description of the threshold of clinical changes in WSSV-infected shrimp, respectively.
It may be necessary to check whether the figures in Figure 5 and Table 3 for threshold of clinical changes in WSSV-infected shrimp are consistent.
line:454
・ Mortality of recipient and donor at 30℃ and 20℃ in the 10^3 group are different between figures and text.
Author Response
Animals
Manuscript ID: ISSN 2076-2615
Title: Evaluation of the Horizontal Transmission of White Spot Syndrome Virus Based on the Disease Severity Grade and Viral Shedding Rate
# Reviewer 2
- First, I will note what I noticed in the argument leading to the conclusion.
Q1. Although the results were obtained in a laboratory setting, the conclusions are reminiscent of the natural occurrence of the pathogen through water in an aquaculture farm. This experiment does not represent a final conclusion on the pathogenicity of WSSV. Therefore, the results are based on the assumption that the virus tested is 100% active. It would be difficult to obtain analytical results if there were conditions that would reduce titer, such as frozen storage.
A1. Thanks for your insights. As you know, the viability of the virus cannot be determined via real-time PCR (genome copies), and it can influence pathogenicity. Therefore, we reported that all experimental infection challenges were carried out using virus that was activated via injection into naïve shrimp within 7 days of the start point of the experiment. Furthermore, to ensure the viability of the virus in this study, we additionally described this content from line 90-92.
Q2. We hope that in the future it will be possible to determine the risk of infection by quantifying the active virus, although this is not necessarily the case with the paper submitted here.
A2. Thank you for your comment. We are aware of the importance of analyzing viral viability, and we acknowledge that viability PCRs (vPCRs) using dyes such as PMA or EMA have been applied to aquatic animal pathogens (Line 578-582). However, the efficiency of vPCR varies depending on the characteristics of the pathogen, such as envelope, genome type, and conditions of PCR assays. Establishing the optimal conditions for WSSV would require further time and effort. Therefore, we have already mentioned the need for vPCR in the Discussion section, and if possible, we plan to establish optimal conditions for analyzing the viability of WSSV in future studies.
Q3. Also, the effect of cannibalism of eating dead shrimp was not mentioned, but even if authors are in a position to discuss only waterborne infections, it needs to be included in the discussion.
A3. Thanks for your comment. Although we agree with your suggestion, we are concerned that the inclusion of a discussion of cannibalism weakens what we are trying to say about this study and may confuse readers. Therefore, we additionally described the influence of cannibalism on horizontal transmission and the requirement of further study including the complex interactions among hosts, vectors, and reservoirs in the Introduction and Discussion. (line 661-664)
Q4. In the experiment in Figure 4, the weight of the shrimp appears to be relevant, but there is no mention of the weight of the shrimp tested. In the series of experiments, the size of the shrimp (weight, age in days, etc.) seems to be an important factor, but it is not mentioned. Were the series of experiments conducted with shrimp of similar size?
A4. Thank you for your comment. We agree with your suggestion and would like to clarify that we used shrimps of similar size throughout the experiment, which is described in section 2.1 of the manuscript. (line 82-83). If it would be helpful to readers, we are willing to revise the manuscript to include the size of shrimp used in each experiment.
- Below are some minor revises
line :86
Q5. Should shortly indicate the specific method of acquiring the virus source.
A5. Thanks for your comment. As per your comment, we additionally described about viral purification process (line 87-90).
line:101-102
Q6. It is the weight of the tissue of the pleopod used for the extraction of total DNA, but is it not appropriate to divide it by the amount of the total DNA extracted?
A6. Thanks for your comment. As a preliminary step, we determined the optimal weight of pleopod tissues for DNA extraction before conducting the study. We tested three different weights (50, 25, and 10 mg; in 25 replicates per group) and found that approximately 10 mg of pleopod tissue showed the optimal concentration for real-time assay (25 ng/uL). While we acknowledge that there may be some loss during the DNA extraction process, we have used the same calculation methods in previous studies and the results have been consistent. However, we understand your concern and appreciate your suggestion. Unfortunately, revising the calculation methods at this point would require extensive changes to the data and results section of the paper, which is not feasible within our timeframe. We kindly ask for your understanding regarding this limitation.
line:127
Q7. artificial infection - replace with experimental infection
A7. Thanks for your suggestion. As per your comment, we replaced artificial infection to experimental infection. (Line 136-137)
line:316
Q8. No explanation for "*" in Figure 3.
A8. Thanks for your suggestion. As per your comment, we added the explanation for “*” in Figure 3 legend (line 343-345).
line:346
Q9. It is reasonable to only observe a trend of positive correlation due to the low r-value.
A9. We have added the p-value in the figure to demonstrate statistical significance, as per your suggestion. Regarding the low r2 value of the linear regression, this is likely due to the fact that approximately 20-30 shrimp in each tank showed various viral copies, even though they were in the same tank. We believe this may be due to individual differences in susceptibility to infection or other factors. Nonetheless, the p-value of the regression was lower than 0.001, indicating a statistically significant relationship between the number of infected shrimp and viral copy numbers.
line:379
Q10. line:379 From Figure 5, the cutoff values for viral shedding and cumulative mortality are 3.1 x 103 (103.49) and 8.7 x 104 (104.97) WSSV DNA copies/mg, respectively, but isn't the mortality 10^4.93? It would be easier to understand if more emphasis is placed on the Criterion in the figure.
line:402
Q11. Table 3. It would be easier to understand if the values of 10^3.49 and 10^4.93 or 4.97 are also included in the description of the threshold of clinical changes in WSSV-infected shrimp, respectively. It may be necessary to check whether the figures in Figure 5 and Table 3 for threshold of clinical changes in WSSV-infected shrimp are consistent.
A10-11. Thanks for your response. It is our editing mistake. As per your comment, we have thoroughly reviewed our manuscript and updated the cut-off value of mortality to 8.5 x 104 (104.93) WSSV DNA copies/mg, ensuring consistency throughout the document.
line:454
Q12. line:454 Mortality of recipient and donor at 30℃ and 20℃ in the 10^3 group are different between figures and text.
A12. Thanks for your response. It is our editing mistake. As per your comment, we have reviewed the value exactly (Line 477-480).
